# The Immediate Effect of Hanging Exercise and Muscle Cylinder Exercise on the Angle of Trunk Rotation in Adolescent Idiopathic Scoliosis

**DOI:** 10.3390/healthcare12030305

**Published:** 2024-01-24

**Authors:** Burçin Akçay, Tuğba Kuru Çolak, Adnan Apti, İlker Çolak

**Affiliations:** 1Department of Physiotherapy and Rehabilitation, Faculty of Health Sciences, Bandırma Onyedi Eylül University, Balıkesir 10200, Turkey; 2Department of Physiotherapy and Rehabilitation, Faculty of Health Sciences, Marmara University, Istanbul 34854, Turkey; tugba.colak@marmara.edu.tr; 3Department of Physiotherapy and Rehabilitation, Faculty of Health Sciences, İstanbul Kültür University, Istanbul 34191, Turkey; a.apti@iku.edu.tr; 4VM Medical Park Maltepe, Istanbul 34846, Turkey; ilker.colak@medicalpark.com.tr

**Keywords:** adolescent idiopathic scoliosis, angle of trunk rotation, scoliosis-specific exercises, Schroth

## Abstract

(1) Background: Semi-hanging and muscle cylinder exercises have been defined as scoliosis-specific corrective exercises. The aim of this study was to evaluate the immediate effect of muscle cylinder and semi-hanging exercises on the angle of trunk rotation in patients with adolescent idiopathic scoliosis (AIS). (2) Methods: Twenty-seven patients with AIS with a mean age of 18.6 years were retrospectively analyzed. The angle of trunk rotation (ATR) values were measured before and after performing semi-hanging and standing muscle cylinder exercises. Both exercises were performed for three to five respiratory cycles. The semi-hanging exercise was performed first, followed by the muscle cylinder exercise, in this order, in all participants. For statistical analysis, the Wilcoxon signed-rank test was used to analyze ATR changes after the exercises, and the Kruskal–Wallis test was used to compare ATR changes according to the main curve location. (3) Results: The thoracic, thoracolumbar and lumbar maximum ATR values were significantly increased after the semi-hanging exercise (*p* < 0.001) and decreased after the muscle cylinder exercise (*p* < 0.001). The ATR change was greater in the lumbar region than in the thoracic and thoracolumbar regions. (4) Conclusion: The results of this study of a small group of patients emphasized that one of the scoliosis-specific corrective exercises, the standing muscle cylinder exercise, improved ATR, while the other, the semi-hanging exercise, worsened ATR in patients with AIS. It is recommended that each scoliosis-specific corrective exercise be evaluated and redesigned to maximize the three-dimensional corrective effect, considering the biomechanics of the spine and the pathomechanics of scoliosis.

## 1. Introduction

Scoliosis involves changes in the spine in three axes and three planes. Lateral flexion occurs in the frontal plane, rotation in the transverse plane and flexion and extension in the sagittal plane, affecting physiological kyphosis and lordosis [1,2,3,4].

Although scoliosis may be a symptom of a disease, it mainly occurs idiopathically in adolescence. Currently, observation, exercise, bracing and surgical interventions are used to treat adolescent idiopathic scoliosis [3]. However, observation should take place via a wait-and-see approach in the teenage period during which rapid growth occurs [5]. Exercises in the conservative treatment of scoliosis can be categorized under two headings: scoliosis-specific exercises and traditional exercises such as stretching and strengthening [2,3,5,6]. Among the scoliosis-specific exercises, the Schroth method is the most studied program and the most effective according to the literature [6,7,8]. In the historical development process, the Schroth method was updated and started to be applied under the name the Schroth Best Practice method in light of evidence-based information. The Schroth Best Practice approach was developed by H.R. Weiss, grandson of Katharina Schroth, the founder of the Schroth method, and son of Christa Lehnert-Schroth, the method’s further developer [2,3,9]. The Schroth Best Practice (SBP) method consists of a six-module treatment program, and one of the modules is based on the Schroth exercises called “Power Schroth”. According to the literature, Power Schroth exercises consist of the five most effective exercises according to the literature, covering highly corrective exercises and core stabilization exercises as well. These five exercises in the Power Schroth module of the Schroth Best Practice method are the “The Musclecylinder, The 50x Exercise, The door handle exercise, The frog at the pond, and Raising the Pelvis”. The muscle cylinder exercise is one of the exercises defined in the Schroth method, and can be performed in side-lying, kneeling and standing positions. This exercise provides unilateral activation of the back muscles specific to scoliotic curvature. The activation of these muscles results in derotation of the lumbar concave transverse processes by changing the orientation of the pelvis, correction in the lumbar region and derotation of the rib hump through force applied to the thoracic convex side [3]. In addition, EMG studies by Weiss et al. have shown that this exercise increases intrinsic lumbar core muscle activity as well as thoracic autochthonous activity when performed standing [10,11,12,13]. The standing muscle cylinder exercise is applied within the Schroth Best Practice program because it is more effective, easier for patients to learn and perform and reliably activates the target muscles already when patients take the starting position [10,11,12,13].

Hanging exercises are also defined in the original and the Intermediate Schroth exercise programs and are widely used by physiotherapists worldwide [3]. Hanging exercises continue to be applied, such as “The Circle” and “Semi-hanging”, within the ISST (International Schroth three-dimensional Scoliosis Therapy^®^) method and BSPTS method (Barcelona Scoliosis Physical Therapy School) [14,15], but these approaches do not involve the application of the Schroth Best Practice method. In the semi-hanging exercise, with the hands on a wall bar above the head, pelvic orientation is achieved with the pelvic corrections described by the Intermediate Schroth method. According to the principles of correction, lateral, dorsal and caudal correction of the pelvis are achieved, expanding all concave parts of the trunk and correcting them through rotational breathing with each rotation [3,14,15].

Semi-hanging and muscle cylinder exercises have been defined as scoliosis-specific corrective exercises, but according to our clinical impressions, it was observed that the rotation component could not be controlled in semi-hanging exercises. In this context, when the literature was checked, there were studies in which all exercises were applied and their effects were examined [6,7,8]. Still, there were no studies in which the effectiveness of different exercises was analyzed separately. Thus, the aim of this study was to evaluate the immediate effect of muscle cylinder and semi-hanging exercises on the angle of trunk rotation. The first hypothesis of this study is that muscle cylinder and semi-hanging exercises have an immediate effect on the angle of trunk rotation. The second hypothesis is that muscle cylinder and semi-hanging exercises have varying effects on trunk rotation.

## 2. Materials and Methods

### 2.1. Study Design and Participants

This retrospective cohort study included data obtained from evaluating adolescents with idiopathic scoliosis who presented at VM Medical Park Hospital between March 2022 and July 2022. The study was approved by the Regional Ethics Committee. All participants and their parents provided written and verbal consent.

Age > 10 years and an AIS diagnosis with a Cobb angle > 10° were the requirements for study inclusion. Patients with non-idiopathic scoliosis, any exercise-related contraindications, a history of spinal surgery, any congenital deformity, any trauma within the previous six months or any accompanying neurological, rheumatological or mental issues were excluded from the study.

### 2.2. Outcomes

Age, gender, body weight, height, current therapies, use of braces, degree of curvature, angle of trunk rotation and pattern of scoliosis were among the demographic and clinical characteristics of the participants that were noted.

A total of 27 participants (23 female, 4 male) with a mean age of 18.6 years were included in the study (Table 1). Eleven participants had a major curve in the thoracic region, seven participants in the thoracolumbar region and nine participants in the lumbar region.

The participants’ type of curvature was determined using the Augmented Lehnert-Schroth (ALS) classification [3]. The validity and reliability of this classification method were demonstrated by Akçay et al. [16].

The curvatures according to the ALS classification are 3CH (functional 3-curve, with hip prominence), 3CTL (functional 3-curve, thoracolumbar with hip prominence), 3CN (functional 3-curve, neutral with a balanced pelvis), 3CL (functional 3-curve with a long lumbar counter curve), 4C (functional 4-curve, double major), 4CL (functional 4-curve with single lumbar) and 4CTL (functional 4-curve with single thoracolumbar curve) [1,3,16]. The functional three-curve patterns define primarily thoracic curves, and the functional four-curve patterns represent double major or single lumbar and thoracolumbar curves with additional lumbosacral curves [1,3,16].

According to the augmented Lehnert-Schroth (ALS) classification, 7 (25.9%) participants had the 4CL curve pattern, 5 (18.5%) participants had the 3CN curve pattern, 4 (14.8%) participants had the 3CL curve pattern, 4 (14.8%) participants had the 4C curve pattern, 4 (14.8%) participants had the 4CTL curve pattern and 3 (11.1%) participants had the 3CTL curve pattern.

The Cobb method, still the gold standard [17], was used to assess the degree of curvature in the coronal plane radiographically. The Cobb angle was manually evaluated by an experienced orthopedist on X-rays taken within the last six months. The Cobb angle was determined by measuring the angle between the perpendiculars drawn from the upper edge of the first vertebra involved in the curvature and the lower edge of the last vertebra on the patient’s X-ray [17].

The most common technique for evaluating scoliosis in clinical settings is the angle of trunk rotation (*ATR*), which measures the intensity of trunk rotation. There is a moderate correlation between the Cobb angle and trunk rotation degree. Without a radiographic evaluation, ATR can be used to monitor and control treatment results [18,19].

To determine the angle of trunk rotation, the patient was instructed to bend forward and let their arms hang loose. A Scoliometer^®^ was positioned on the patient’s back, and the maximum degree was noted separately for the thoracic, thoracolumbar, lumbar and sacral regions [20,21]. ATR measurements were repeated before and after each exercise.

### 2.3. Interventions

ATR measurements were performed and recorded before the patients started the exercises. Then, both exercises were performed for 3 to 5 respiratory cycles, and the ATR measurement was repeated. Firstly, a semi-hanging exercise was performed, and measurements were taken; then, a muscle cylinder exercise was performed, and this was performed in this order in all participants. The patient rested seated for 5 min between each exercise. ATR values were measured by an experienced physiotherapist in three repetitions, and mean values were used. An experienced physiotherapist with Schroth Best Practice and the Intermediate Schroth certification supervised all exercises.

Hanging exercises involve elongation and mobilization of the spine in the midline or the patient’s position, which can be changed according to the curve [1]. In the Schroth Best Practice approach, hanging exercises are not applied in mild and moderate curvatures, as they are thought to lead to a further decrease in sagittal plane curvatures [2,3]. However, in earlier years, hanging exercises were used in the original and intermediate Schroth period [1,14,15]. The original Schroth method was defined, and the semi-hanging exercise for severe curvatures was developed. In severe scoliotic curves, the thoracic region behaves as kyphotic in the sagittal plane due to increased severe gibbosity [1] (Figure 1b).

The standing muscle cylinder exercise addresses the autochthonous back musculature unilaterally: in the lumbar region, it addresses the intrinsic lumbar part that is characterized by an oblique orientation from the pelvis up to the transverse processes. Correction of the spine is achieved simultaneously in the frontal, sagittal and horizontal planes [2,3] (Figure 1a).

### 2.4. Statistical Methods

The data obtained in this study were analyzed statistically using IBM SPSS Statistics version 11 software (IBM Corporation, Armonk, NY, USA). The normality of the data was analyzed with the Shapiro–Wilk test. Descriptive statistics were reported as mean, median, standard deviation, minimum and maximum values. The Wilcoxon signed-rank test was used to analyze ATR changes after the exercises. The Kruskal–Wallis test was used to compare ATR changes after the exercises according to major curve location. Spearman correlation analysis was used for determining the correlation between the Cobb angle and ATR changes before and after exercise. A value of *p* < 0.05 was deemed statistically significant for all tests.

## 3. Results

The maximum ATR value before the exercises was 8.8°, the maximum ATR value obtained after the semi-hanging exercises was 10.2° and the maximum ATR value measured after the muscle cylinder exercises was 7.2°. It was determined that the ATR values after the semi-hanging exercise were statistically higher than the initial values (*p* < 0.001) and after the muscle cylinder exercise, the values (*p* < 0.001) were significantly lower than the initial values (Figure 2).

When the changes produced by the exercises in the ATR measurements were analyzed according to the region of the major curvature, it was determined that there was an increase in ATR measurements of all thoracic, thoracolumbar and lumbar regions after the semi-hanging exercise, and this increase was the highest in the lumbar region (1.5° ± 1.2). After the standing muscle cylinder exercise, it was determined that all ATR values obtained in the thoracic, thoracolumbar and lumbar regions decreased compared to the baseline, and this decrease was the highest in the lumbar region (−2.1 ± 1.4) (Figure 2, Table 2). However, the change in ATR according to the location of the apex was statistically similar for the semi-hanging and muscle cylinder exercises (*p* = 0.879, *p* = 0.943).

The Spearman correlation analysis showed that the changes in ATR values after the exercises were not related to the maximum Cobb values (*p* = 0.277, r = −0.217), whereas the changes in the maximum ATR values after the muscle cylinder exercises were negatively and moderately related to the maximum ATR angle values (*p* = 0.034, r = −0.409).

## 4. Discussion

This study revealed that two different exercises recommended specifically for scoliosis may have different effects on curvatures. The results of the current study showed that the ATR values were significantly reduced after the muscle cylinder exercise; on the contrary, there was a significant increase after the semi-hanging exercise. In addition, ATR changes were recorded in the lumbar region at the highest rate in both exercises, and these changes were not related to the severity of the Cobb angle and apex of curvature.

Six different scoliosis-specific exercise approaches are applied to scoliosis patients by physiotherapists worldwide [22]. The Schroth method is the most widely used and has been included in most scientific studies [8]. The Schroth method is a scoliosis-specific rehabilitation method developed by Katharina Schroth in 1921 for thoracic curves with a Cobb angle exceeding 70 to 80 degrees [1,3]. The method has continued to be developed and updated by her daughter, Christa Lehnert-Schroth, PT, and grandson, Dr. Hans R. Weiss, for three generations [3,9]. Today, the Schroth method is practiced using three concepts with common fundamental principles. These include the Barcelona Scoliosis Physical Therapy School (BSPTS) led by Dr. Rigo in Spain; International Schroth 3D Scoliosis Therapy (ISST) led by Axel Hennes, the head physiotherapist who retired from Asklepios Katharina Schroth Clinic, Germany; and the Schroth Best Practice (SBP) led by Dr. Weiss [1,3,8,9,22]. The main difference between the Schroth Best Practice approach and the other two Schroth-based approaches is maintaining physiological sagittal plane curvatures during all exercises, ensuring their maintenance in daily living activities and performing overcorrection instead of midline correction in frontal plane corrections. In line with these principles, hanging exercises have not been implemented in the Schroth Best Practice method [2,3].

In this study, it was found that the ATR values improved significantly after the muscle cylinder exercises in all patients, but the ATR values worsened after the semi-hanging exercises. Regarding the clinical impression of the authors, it was observed that it was easier for AIS patients to maintain physiological curves in the sagittal plane during the muscle cylinder exercises, while it was difficult for them to keep physiological curves in the semi-hanging exercises and make frontal plane corrections simultaneously. The difference in the effects of these two corrective exercises may be attributed to the inability to maintain the correction of sagittal curvatures. However, the sagittal profile is important in adolescent idiopathic scoliosis, and flat back syndrome is highly prevalent [23,24,25]. It was first mentioned in 1925 by Farkas that thoracic flatback may be a trigger factor in the development of scoliosis [26]. To obtain a correction effect in three planes, Weiss et al. applied a Chêneau-style brace and a “sagittal counterforce” brace, which was intended to correct the sagittal plane. Improvements were seen in the frontal and coronal planes with both brace types, and the authors suggested that sagittal forces could be applied to produce a three-dimensional improvement [27]. Kuru Çolak et al. demonstrated that using sagittal forces can reduce the vertebral rotation angle and improve spine flexibility. Reducing vertebral rotation and increasing spine flexibility may prevent scoliosis progression. Symmetric mobilization exercises ought to be included in the AIS rehabilitation program [28]. While most physical therapy programs specifically for scoliosis focus on correcting lateral deformity, some also aim to correct rotation, and very few address the sagittal profile [1,2,3,22,27,28,29,30].

According to the Schroth Best Practice principles, addressing physiological lumbar lordosis and thoracic kyphosis in the sagittal plane is essential for correcting scoliotic curves with the help of exercise. The sagittal profile can be provided with shoulder girdle corrections while sitting and standing [2,3]. Bilateral and unilateral arm elevations have been indicated to lead to thoracic spine extension [31,32,33,34]. In addition to these studies, Edmondston et al. evaluated the effect of bilateral arm elevation on the extension of the thoracic region more objectively using radiographic and photographic methods and concluded that bilateral arm elevation requires an extension movement close to the full range of motion of the thoracic spine. It was stated that with bilateral shoulder elevation, thoracic extension motion was greater in the lower thoracic region than in the upper thoracic region [35]. Also, in 1953, Keegan evaluated the relationship between the change in the angle between the thigh–trunk angle and lordosis with radiography. He showed that when the thigh–trunk angle was decreased from 200 degrees to 50 degrees (increasing the degree of hip flexion), the pelvis was displaced posteriorly, and lumbar lordosis decreased. He stated that this change in the pelvis may be due to muscle tension depending on the position. He suggested that psoas tension acts to increase lumbar lordosis while standing, that a thigh–trunk angle of 135 degrees is a neutral position and that tension in the posterior thigh muscles plays a major role in the posterior rotation of the pelvis and the reduction in lumbar lordosis, as well as in the lever arm for posterior pelvic rotation caused by sitting on the ischial tuberosities when this degree is exceeded [36,37]. The arm elevation and increasing hip flexion degree that occur when performing the semi-hanging exercise may also lead to a further reduction in thoracic kyphosis and lumbar lordosis, which is already reduced in AIS, and therefore, no improvement in ATR may have been observed after the semi-hanging exercise. Although the patient was encouraged to maintain the physiological sagittal profile during the semi-hanging exercises, this may not have been achieved due to the lordosis and kyphosis-reducing effect of biomechanical forces. According to biomechanical principles, lateral flexion–rotation and flexion or extension movements occur together in the spine [38]. It has also been shown in the literature that ATR values are correlated with the Cobb angle [18,19]. In light of this information, each exercise that addresses scoliosis-specific three-dimensional correction principles should be considered in the context of scoliosis pathomechanics and spinal biomechanics.

Lovett described the complex movement patterns of the spine very early on at the beginning of the 20th century. He showed that a torso movement in only one plane is accompanied by a movement in the other two planes, i.e., that any movement of the spine must be a three-dimensional movement. In the case of scoliosis correction, this means the following: if the pathological spinal misalignment is reinforced in one plane (and this occurs with the flat back adjustment during the semi-hanging exercise), the correction of the spinal misalignment in the other two planes is blocked [39,40]. This may explain the worsening of the ATR when using semi-hanging exercises.

In the present study, ATR changes, including a decrease in the muscle cylinder exercise and an increase in the semi-hanging exercise, were obtained in the lumbar region to a greater extent than in the thoracic and thoracolumbar regions in both exercises. The fact that the lumbar region was affected more than other regions may be because the lumbar region has a larger normal range of motion. While the cervical and lumbar spine are stabilized only by muscles, the thoracic spine is additionally stabilized by the rib cage, ribs and costovertebral ligaments. It has been shown that all thoracic cage structures contribute to the stability of the thoracic spine [41]. While this plays an important role in compressive loads [42], it also provides stiffness in the thoracic spine by limiting flexibility, especially axial rotation [41]. In addition, the results of this study may also indicate the effectiveness of lumbar physiological lordosis in correcting and maintaining the sagittal profile. In future studies, planning a study in which the amount of lumbar lordosis is measured during exercise may clarify the effect of lumbar physiological lordosis for curvature correction. It may also imply that lumbar pelvic corrections can be performed more effectively in the muscular cylinder exercise but not effectively in the semi-hanging exercise. Internal and external forces acting on the spine, such as muscle activation and spinal biomechanics, need to be evaluated in further studies for semi-hanging exercises.

The results of our study also call into question the current SOSORT definition of scoliosis-specific exercises (PSSEs). PSSE includes all exercises described in the literature on scoliosis treatment, regardless of any evidence of effectiveness. A large number of treatment programs are not specific as the corrections are not carried out according to the patient’s individual curvature pattern. At present, this is only the case for the treatment programs based on the Monticone and the Schroth derivatives [5,43].

We therefore suggest that the pattern-specific programs be called Pattern-Specific Scoliosis Rehabilitation (PSR). Within this group, the effectiveness of individual exercises should also be investigated further.

In the current study, exercise-induced ATR changes were unrelated to the Cobb angle’s severity and the curve’s apex. The corrective effect of exercise designed according to the curve pattern may be greater on ATR change than the severity of the Cobb angle and the curve’s apex. However, long-term results are needed to confirm these findings.

### Limitations and Further Implications

To the best of our knowledge, no previous studies have evaluated the effects of different exercises one-by-one on scoliotic deformity. Studies generally evaluate the effectiveness of an exercise program or brace treatment combined with exercise after a certain period. In this sense, our study will shed light on the literature and the methods of clinicians.

This study has some limitations. All patients performed semi-hanging, and then, muscle cylinder exercises; the effect of the muscle cylinder exercises before the semi-hanging exercises could not be analyzed. Performing exercises in a randomized order and evaluating their effects may be recommended, but this would require a study design with the patients matched to gender, age, curve pattern and Cobb angle (matched pairs). The long-term effects of these exercises are yet to be evaluated. Since there was no long-term follow-up evaluation, it was impossible to evaluate the effect of these two exercises on the Cobb angle, which is evaluated in the frontal plane and is still the gold standard. Future studies should investigate how different exercises affect scoliosis, a three-dimensional deformity with different curvature patterns and degrees. This study is a preliminary study carried out with a small group of patients to demonstrate an effect that was noticed through clinical observation. In this study, patients with and without complete skeletal maturation were included. In future studies, it may be recommended to evaluate scoliosis patients by classifying them into different groups according to skeletal maturation. Also, kinetic and kinematic analysis (including the frontal, sagittal and coronal planes) measurements, including all extremities and the spine, during the performance of all exercises, and even electromyography measurements evaluating muscle activation, can be performed in further studies with a larger number of patients and long-term evaluations. In addition, studies on the biomechanical aspects of scoliosis-specific exercises are limited and should be conducted to allow patients to participate in more effective exercises in the clinic.

## 5. Conclusions

The results of the present study of a small group of patients emphasized that muscle cylinder exercise, one of the scoliosis-specific corrective exercises, improved the angle of trunk rotation, and the semi-hanging exercise worsened the angle of trunk rotation in patients with AIS. The change in trunk rotation was greater in the lumbar region than in other spinal regions in both exercises. Professionals working in the scoliosis specialty know that there is a limited time between the onset of pubertal growth and its deceleration, especially given the rapid progression of scoliosis during the age of rapid growth. Not losing time due to improper exercise techniques or treatment modalities is very important. Thus, it is recommended that each scoliosis-specific corrective exercise should be evaluated and redesigned to maximize the three-dimensional corrective effect, considering the biomechanics of the spine and the pathomechanics of scoliosis.

## Figures and Tables

**Figure 1 healthcare-12-00305-f001:**
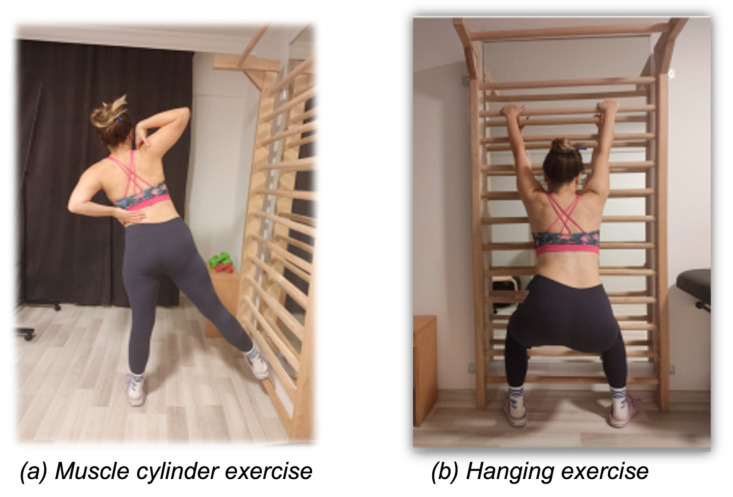
Examples of (**a**) muscle cylinder and (**b**) hanging exercises.

**Figure 2 healthcare-12-00305-f002:**
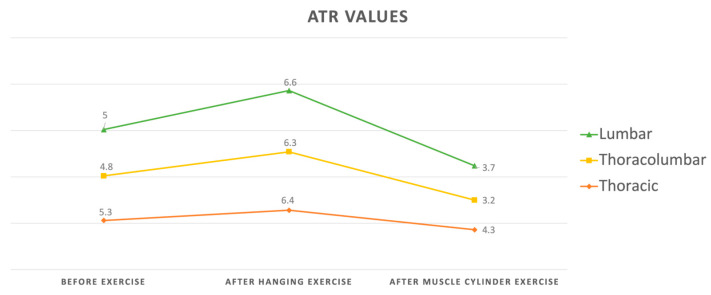
Changes in mean ATR values with semi-hanging and muscle cylinder exercises.

**Table 1 healthcare-12-00305-t001:** Demographic and clinical characteristics of the patients.

	Mean ± SD Median (Min–Max)
Age (years)	18.6 ± 4.31 18 (13–25)
Height (cm)	164.7 ± 11.37 164 (144–188)
Weight (kg)	56.22 ± 15.2 52 (40–100)
BMI (kg/m^2^)	20.55 ± 2.9 20 (15.7–28.2)
Maximum Cobb angle (°)	33.9 ± 7.3 30 (18–50)
Maximum ATR angle (°)	8.8 ±5.3 8 (1–23)

cm: centimeter, kg: kilogram, kg/m^2^: kilogram per square meter, °: degree, ATR: angle of trunk rotation, SD: standard deviation, Min: minimum, Max: maximum.

**Table 2 healthcare-12-00305-t002:** Changes in ATR values after semi-hanging and muscle cylinder exercises.

ATR Values	Before Mean ± SD	After Semi-Hanging Exercise Mean ± SD	After Muscle Cylinder Exercise Mean ± SD	*p* Value
Thoracic	5.37 ± 6.19	6.44 ±6.29	4.31 ± 6.07	<0.001
Thoracolumbar	4.83 ± 3.68	6.31 ± 4.05	3.22 ± 2.80	<0.001
Lumbar	5.01 ± 3.55	6.62 ± 4.55	3.70 ± 3.33	<0.001

ATR: angle of trunk rotation, SD: standard deviation.

## Data Availability

The data presented in this study are available on request from the corresponding author. The data are not publicly available as data collection is ongoing.

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
