# Peer review of "The Immediate Effect of Hanging Exercise and Muscle Cylinder Exercise on the Angle of Trunk Rotation in Adolescent Idiopathic Scoliosis"

_healthcare, 2024, doi:10.3390/healthcare12030305_

Round 1
Reviewer 1 Report
Comments and Suggestions for Authors
The paper concerns an important problem of the rehabilitation effectiveness in scoliosis.
Introduction is well written, the goal of the study is clearly stated.
Material & Methods: patients with mature spine (Risser sign >4) should not be included, they do not need treatment below 40 deg acc to Cobb and they need surgery when scoliosis is bigger. Classic indications for exercises are Cobb angle up to 40 (max) in growing spine.
Results should be analyzed with some follow up period and with Cobb angle as well.
Figure 2 and table 2 - I did not find them.
To sum up, I think that problem is important, but the younger patients (without skeletal maturity should be included) and assessment in two planes (sagittal and coronal) should be performed.
Comments on the Quality of English Languagethere are some problems with understanding the material/ methods and results (style)
Author Response
Reviewer 1
Comments to the Author:
The paper concerns an important problem of the rehabilitation effectiveness in scoliosis.
Introduction is well written, the goal of the study is clearly stated.
Response: We thank you very much for the comments and suggestions. The comments and suggestions are valuable and helpful for revising and improving our manuscript. We have made revisions according to your comments and suggestions and yellow-highlighted them in the manuscript. With your guidance, this study will make valuable contributions to the literature and clinical practice.
Thank you very much.
Sincerely
1) Material & Methods: patients with mature spine (Risser sign >4) should not be included, they do not need treatment below 40 deg acc to Cobb and they need surgery when scoliosis is bigger. Classic indications for exercises are Cobb angle up to 40 (max) in growing spine.
Response: Considering the indications for conservative treatment guidelines, patients with indications for exercise were included in the study.
Weiss, H. R., Negrini, S., Rigo, M., Kotwicki, T., Hawes, M. C., Grivas, T. B., Maruyama, T., & Landauer, F. (2008). Indications for conservative management of scoliosis (SOSORT guidelines). Studies in health technology and informatics, 135, 164–170.
Dereli, E. E., Gong, S., Çolak, T. K., & Turnbull, D. (2021). Guidelines for the conservative treatment of spinal deformities–Questionnaire for a Delphi consensus. The South African journal of physiotherapy, 77(2).
2) Results should be analyzed with some follow up period and with Cobb angle as well.
Response: You are very accurate, it would be important to give the follow-ups of the Cobb angle. However, in this study, the immediate effect is examined. Your suggestion has been added to the limitation section for further studies.
3) Figure 2 and table 2 - I did not find them.
Response: Table 2 and figure 2 were forgotten to be deleted on templete and revised.
4) To sum up, I think that problem is important, but the younger patients (without skeletal maturity should be included) and assessment in two planes (sagittal and coronal) should be performed.
Response: Thank you very much your suggestions. Your suggestion has been added to the limitation section for further studies.
5) English language editing was done by MPDI editing service.

Reviewer 2 Report
Comments and Suggestions for Authors
Introduction
The presentation of Schroth's schools can be confusing for the reader. The "semi hanging" exercise is a geometric detorsion, while the "circle" is a mechanical detorsion. More interesting could be the biomechanical approach outlined in the conclusion (line 310). Below 40°, mechanical detorsion is more effective (Panjabi), which explains the results obtained in this work for scoliosis averaging 34°.
METHODS
The duration, intensity and type of muscle contraction could be clarified.
DISCUSSION
The discussion of the sagittal plane in relation to coupled movements is also a source of confusion for the Schroth method. Restoring physiological isostatic balance according to pelvic incidence improves mobility in the frontal plane and derotation. Is this how the exercises were carried out?
Schroth is currently a generic term for "PSSE". The general impression is one of confusion accentuated by the complexity of the classification, while the results have the merit of emphasizing the value of mechanical detorsion for scoliosis of less than 40°.
Minor precisions
line 253: It has also been shown in the literature that ATR values are correlated with Cobb angle. Can you add the reference?
line 64 according to "your" clinical impressions, Are you sure?
Author Response
Reviewer 2
We thank you very much for the comments and suggestions. The comments and suggestions are valuable and helpful for revising and improving our manuscript. We have made revisions according to your comments and suggestions and green-highlighted them in the manuscript. With your guidance, this study will make valuable contributions to the literature and clinical practice.
Thank you very much.
Sincerely
Comments to the Author:
Introduction
1) The presentation of Schroth's schools can be confusing for the reader. The "semi hanging" exercise is a geometric detorsion, while the "circle" is a mechanical detorsion. More interesting could be the biomechanical approach outlined in the conclusion (line 310). Below 40°, mechanical detorsion is more effective (Panjabi), which explains the results obtained in this work for scoliosis averaging 34°.
Response: Thank you very much for your suggestion. However, we were unable to find a reference to this information. Only one study on bracing uses these terms, but no study on exercise uses these terms.
de Mauroy, J. C., Lecante, C., Barral, F., & Pourret, S. (2014). Prospective study and new concepts based on scoliosis detorsion of the first 225 early in-brace radiological results with the new Lyon brace: ARTbrace. Scoliosis, 9(1), 1-19.
METHODS
2) The duration, intensity and type of muscle contraction could be clarified.
Response: Isometric muscle contraction was requested at a dose that would not disturb the corrected position in the starting position and maintain the corrected position of exercise. Also, 3 to 5 cycles of rotational breathing were requested. Each respiration included deep inspiration and expiration. The exercise was interrupted when it was felt that the correction was impaired.
DISCUSSION
3) The discussion of the sagittal plane in relation to coupled movements is also a source of confusion for the Schroth method. Restoring physiological isostatic balance according to pelvic incidence improves mobility in the frontal plane and derotation. Is this how the exercises were carried out?
Response: The pelvic corrections in the exercises were achieved in the starting position and maintained throughout the exercise. A detailed explanation about this point has been added to the introduction section.
4) Schroth is currently a generic term for "PSSE". The general impression is one of confusion accentuated by the complexity of the classification, while the results have the merit of emphasizing the value of mechanical detorsion for scoliosis of less than 40°.
Response: Schroth exercises are differentiated from other PSSE methods as the most widely used and evidence-based exercise method by physiotherapists. Therefore the term Schroth is used.
Minor precisions
line 253: It has also been shown in the literature that ATR values are correlated with Cobb angle. Can you add the reference?
Response: The reference added.
line 64 according to "your" clinical impressions, Are you sure?
Response: Revised.
English language editing was done by MPDI editing service.

Reviewer 3 Report
Comments and Suggestions for Authors
Review healthcare-2782769-peer-review-v1
The aim of the paper The immediate effect of hanging exercise and muscle cylinder exercise on the angle of trunk rotation in adolescent idiopathic scoliosis was to evaluate the immediate effect of two exercises, muscle cylinder, and semi-hanging exercises, on the angle of trunk rotation in patients with adolescent idiopathic scoliosis (AIS). The paper is interesting and quite well written, but it requires some clarifications and additions, especially in the description of the research methodology and results. Below you will find the comments in the order they appear in the text and not in order of importance.
Title and abstract:
1) The word title should be removed from line 2
2) The word these should be removed from line 17.
3) In the abstract, lines 21 and 22 should be deleted, as they do not introduce anything. Please provide numerical values for further description. The abstract in the methods section lacks a description of statistics and a definition of the exercises performed.
Introduction:
1) Line 51 - what five exercises? Please add information about them.
2) Line 52 - what is meant by the muscle cylinder exercise - please add a description. The same applies to lines 59 and 63, where you need to add a definition/ description of hanging exercises and semi-hanging exercises.
Material and methods:
1) In section 2.1, please put the information that are in lines 141 - 151. In addition, in line 148, please add a citation. Lines 146 - 147 can be removed, as these are typical labels only the description for ATR should remain. I see that the description is in lines 117 - 131 - so please move this information.
2) In line 83, there should be a description of the abbreviation ATR, because it appears for the first time in the text (I do not include the abstract, because one may not read it). In line 90 - please discuss how ATR was evaluated. The order of Figure 1a and b should be changed because they are discussed differently in the text. The description for Figure 1 should look like the following after the change: Examples of hanging and muscle cylinder exercises: a) muscle cylinder excercise, b) Haning exercise.
3) The information contained in lines 107 - 109 should be removed. The information from lines 110 – 112 should appear before in the text. I suggest line 82.
Results:
I find this section the weakest and not readable. The authors need to improve the whole section. Below are suggestions.
1) Line 168 - please use Figure instead of Graphic.
2) Graphics should be better elaborated. Standard deviations are missing. I suggest instead of Figure include a table, which will be more readable.
3) Line 164 - I do not understand the result (-2.1 ± 1.4).
4) Line 166 - what are the numbers (p= 879, p= 943)?
5) Lines 169 - 172 - there is no description in the methods that correlation will be used. Please supplement. It is not clear what is correlated with what.
6) Please clarify the information from lines 173 - 181.
Author Response
Reviewer 3
Comments to the Author:
The aim of the paper The immediate effect of hanging exercise and muscle cylinder exercise on the angle of trunk rotation in adolescent idiopathic scoliosis was to evaluate the immediate effect of two exercises, muscle cylinder, and semi-hanging exercises, on the angle of trunk rotation in patients with adolescent idiopathic scoliosis (AIS). The paper is interesting and quite well written, but it requires some clarifications and additions, especially in the description of the research methodology and results. Below you will find the comments in the order they appear in the text and not in order of importance.
Response: We thank you very much for the comments and suggestions. The comments and suggestions are valuable and helpful for revising and improving our manuscript. We have made revisions according to your comments and suggestions and blue-highlighted them in the manuscript. With your guidance, this study will make valuable contributions to the literature and clinical practice.
Thank you very much.
Sincerely
Title and abstract:
1) The word title should be removed from line 2
Response: Removed.
2) The word these should be removed from line 17.
Response: Removed.
3) In the abstract, lines 21 and 22 should be deleted, as they do not introduce anything. Please provide numerical values for further description. The abstract in the methods section lacks a description of statistics and a definition of the exercises performed.
Response: Revised and added in method section as “Both exercises were performed for 3 to 5 respiratory cycles. The semi-hanging exercise was performed first, followed by the muscle cylinder exercise and performed in this order in all participants. For statistical analysis, Wilcoxon signed-rank test was used to analyse ATR changes after the exercises, and Kruskal-Wallis test was used to compare ATR changes according to the main curve location.”
Introduction:
1) Line 51 - what five exercises? Please add information about them.
Response: Added.
2) Line 52 - what is meant by the muscle cylinder exercise - please add a description. The same applies to lines 59 and 63, where you need to add a definition/ description of hanging exercises and semi-hanging exercises.
Response: Added.
Material and methods:
1) In section 2.1, please put the information that are in lines 141 - 151. In addition, in line 148, please add a citation. Lines 146 - 147 can be removed, as these are typical labels only the description for ATR should remain. I see that the description is in lines 117 - 131 - so please move this information.
Response: Revised.
2) In line 83, there should be a description of the abbreviation ATR, because it appears for the first time in the text (I do not include the abstract, because one may not read it). In line 90 - please discuss how ATR was evaluated. The order of Figure 1a and b should be changed because they are discussed differently in the text. The description for Figure 1 should look like the following after the change: Examples of hanging and muscle cylinder exercises: a) muscle cylinder excercise, b) Haning exercise.
Response: Revised.
3) The information contained in lines 107 - 109 should be removed. The information from lines 110 – 112 should appear before in the text. I suggest line 82.
Response: Revised.
Results:
I find this section the weakest and not readable. The authors need to improve the whole section. Below are suggestions.
1) Line 168 - please use Figure instead of Graphic.
Response: Revised.
2) Graphics should be better elaborated. Standard deviations are missing. I suggest instead of Figure include a table, which will be more readable.
Response: Added.
3) Line 164 - I do not understand the result (-2.1 ± 1.4).
Response: The "-" here indicates a decrease.
4) Line 166 - what are the numbers (p= 879, p= 943)?
Response: Revised.
5) Lines 169 - 172 - there is no description in the methods that correlation will be used. Please supplement. It is not clear what is correlated with what.
Response: Added in method section.
6) Please clarify the information from lines 173 - 181.
Response: Revised.
7) English language editing was done by MPDI editing service.

Reviewer 4 Report
Comments and Suggestions for Authors
Interesting article on a previously unexplored topic. The article concerns effect of hanging exercise and muscle cylinder exercise on scoliosis.
My comments:
Abstract.
Well presented.
Introduction:
There are very few articles confirming the effectiveness of hanging exercise and muscle cylinder exercise in the treatment of scoliosis. Therefore, it is necessary to broadly and clearly justify the use of this method and the effectiveness of these exercises.
Please add a broader review of articles on exercises in the treatment of scoliosis, especially articles on the use of hanging exercise and muscle cylinder exercise on scoliosis treatment.
Please add a research hypothesis.
Materials and methods:
Please describe in detail how the Cobb angle was calculated. In what software, based on what X-rays - detailed description.
Statistical methods correct.
Results:
Well presented.
Tables and figures are good and legible.
Discussion:
Please add more work limitations.
Please write that the study group was small - the results are preliminary and cautious.
Please add that studies on larger numbers of patients and long-term evaluations are necessary in the future.
Conclusions:
Please add that all conclusions are based on the results of studies of a small group of patients.
Author Response
Reviewer 4
Comments to the Author:
Interesting article on a previously unexplored topic. The article concerns effect of hanging exercise and muscle cylinder exercise on scoliosis.
Response: We thank you very much for the comments and suggestions. The comments and suggestions are valuable and helpful for revising and improving our manuscript. We have made revisions according to your comments and suggestions and pink-highlighted them in the manuscript. With your guidance, this study will make valuable contributions to the literature and clinical practice.
Thank you very much.
Sincerely
My comments:
1) Abstract.
Well presented.
2) Introduction:
There are very few articles confirming the effectiveness of hanging exercise and muscle cylinder exercise in the treatment of scoliosis. Therefore, it is necessary to broadly and clearly justify the use of this method and the effectiveness of these exercises.
Please add a broader review of articles on exercises in the treatment of scoliosis, especially articles on the use of hanging exercise and muscle cylinder exercise on scoliosis treatment.
Response: There are many articles about Schroth method in the literature and the level of evidence is available. For this reason, all relevant articles in the literature have been included. In these studies, the efficacy of all exercises has been stated and the efficacy of the exercises individually has not been presented.
Please add a research hypothesis.
Response:
Materials and methods:
Please describe in detail how the Cobb angle was calculated. In what software, based on what X-rays - detailed description.
Response: Added.
Statistical methods correct.
Results:
Well presented.
Tables and figures are good and legible.
Discussion:
Please add more work limitations.
Please write that the study group was small - the results are preliminary and cautious.
Please add that studies on larger numbers of patients and long-term evaluations are necessary in the future.
Response: Added.
Conclusions:
Please add that all conclusions are based on the results of studies of a small group of patients.
Response: Added.
English language editing was done by MPDI editing service.

Round 2
Reviewer 1 Report
Comments and Suggestions for Authors
The paper is improved. However, I did not see any about skeletal maturity/ older patients in limitations of the study.
To sum up, the paper (in my opinion) can be published
Author Response
We thank you very much for the comments and suggestions. The comments and suggestions are valuable and helpful for revising and improving our manuscript. We have made revisions according to your comments and suggestions and blue-highlighted them in the manuscript. With your guidance, this study will make valuable contributions to the literature and clinical practice.
In line with your suggestion, an addition has been made to the limitation section.
Thank you very much.
Sincerely
Reviewer 3 Report
Comments and Suggestions for Authors
The authors have complied with my recommendations. I have no additional comments.
Author Response
We thank you very much for the comments and suggestions. The comments and suggestions were valuable and helpful for revising and improving our manuscript. With your guidance, this study will make valuable contributions to the literature and clinical practice.
Thank you very much.
Sincerely
Reviewer 4 Report
Comments and Suggestions for Authors The authors made the suggested changes to the manuscript. Manuscript acceptable.
Author Response
We thank you very much for the comments and suggestions. The comments and suggestions were valuable and helpful for revising and improving our manuscript. With your guidance, this study will make valuable contributions to the literature and clinical practice.
Thank you very much.